# Strangers, Friends, and Lovers Show Different Physiological Synchrony in Different Emotional States

**DOI:** 10.3390/bs10010011

**Published:** 2019-12-22

**Authors:** Andrea Bizzego, Atiqah Azhari, Nicola Campostrini, Anna Truzzi, Li Ying Ng, Giulio Gabrieli, Marc H. Bornstein, Peipei Setoh, Gianluca Esposito

**Affiliations:** 1Department of Psychology and Cognitive Science, University of Trento, 38068 Rovereto TN, Italy; andrea.bizzego@unitn.it (A.B.); n.campostrini@gmail.com (N.C.); 2Psychology Program, School of Social Sciences, Nanyang Technological University, Singapore 639818, Singapore; nura0066@e.ntu.edu.sg (A.A.); LNG030@e.ntu.edu.sg (L.Y.N.); GIULIO001@e.ntu.edu.sg (G.G.); psetoh@ntu.edu.sg (P.S.); 3Trinity College Institute of Neuroscience, Trinity College Dublin, Dublin 2, Ireland; truzzia@tcd.ie; 4National Institute of Child Health and Human Development, Rockville, MD 20847, USA; marc.h.bornstein@gmail.com; 5Institute for Fiscal Studies, London WC1E 7AE, UK

**Keywords:** heart rate variability, dyads, physiological synchrony, relationship, emotion

## Abstract

The mere copresence of another person synchronizes physiological signals, but no study has systematically investigated the effects of the type of emotional state and the type of relationship in eliciting dyadic physiological synchrony. In this study, we investigated the synchrony of pairs of strangers, companions, and romantic partners while watching a series of video clips designed to elicit different emotions. Maximal cross-correlation of heart rate variability (HRV) was used to quantify dyadic synchrony. The findings suggest that an existing social relationship might reduce the predisposition to conform one’s autonomic responses to a friend or romantic partner during social situations that do not require direct interaction.

## 1. Introduction

As social mammals, humans need to bond with others in order to initiate and maintain social relationships. Fostering of affiliation in a shared environment requires the alignment of higher-order emotional states of both members of the dyad. Synchrony, the mutual attunement and adaptation of these emotional states between partners [1], can be reflected at the physiological level in the parameters of the autonomic nervous system (ANS) [2,3]. For instance, Reference [4] showed that simply imitating the facial expressions of a randomly paired stranger increased the synchrony of heart rate patterns in both members of a dyad. Autonomic synchrony in pairs of strangers was also correlated with an enhanced ability to complete collaborative tasks [4,5]. The primary function of the ANS and its two divisions (i.e., parasympathetic and sympathetic branches) is to maintain homeostasis in the individual. Therefore, a shift of the ANS reflects a need to adjust physiological levels of arousal so that homeostasis continues to be supported. Emotional arousal like alertness and excitement in response to social stress or novelty is linked to the activation of the sympathetic branch [6], whereas a relaxed emotional condition that premeditates attentional focus is associated with the calming response of the parasympathetic branch [7,8]. The involvement of the sympathetic and parasympathetic branches thus allows for interpretation of the physiological synchrony experienced by dyadic partners. Synchronous autonomic arousal has been observed in strangers who shared the same immediate social environment [9,10]. Golland and colleagues [9] demonstrated that the mere copresence of a stranger in the absence of direct face-to-face communication was sufficient to establish physiological synchrony at the autonomic level. In their study, participants who were strangers sat next to each other and watched a series of movie clips that elicited different emotional states. Golland and colleagues [9] collected an index of autonomic arousal by measuring the inter-beat interval (IBI) between two successive heart periods during each emotional state before computing the physiological synchrony attained between participants. Subjective self-reported emotional responses were recorded from the participants as well. Comparing physiological synchrony of autonomic signals with emotional responses, the authors found that the degree to which autonomic arousal was synchronised correlated with the similarity of emotional experiences reported by the strangers. This revelation indicates that emotional affiliation between copresent strangers may be mediated primarily by physiological synchrony. While Reference [9] has demonstrated how the mere copresence of strangers establishes physiological synchrony, the effect of copresence has not been investigated among dyads in other relational categories, such as between pairs of friends and romantic partners. In a lifetime, individuals find themselves differentially affiliated to several others within social pair-bonds. During the establishment of these selective attachments, a biobehavioral reorganization is thought to occur in which multiple biological, behavioral, and cognitive processes between partners come to coincide [11]. Through repeated interactions, partners become increasingly sensitized to one another’s unique rhythms and cues, which, over time, become ingrained and reflected at a physiological level. In romantic pairs, synchronised heart rate variability (HRV) emerged in laboratory tasks [12], while within-couple hormonal associations have been shown to predict levels of empathy [13] and connectedness [14]. Given the likelihood that physiological synchrony has been previously established in dyads that share greater emotional closeness (i.e., friends and romantic partners), the mere copresence of the partner might elicit a different and possibly stronger physiological synchrony compared to dyadic strangers. The function of physiological synchrony in positive as opposed to negative emotional states remains uncertain. There is naivete in the assumption that synchrony is only commonly associated with positive emotional states [11,15,16,17,18] as numerous contradictory findings challenge the generality of this principle [19,20,21,22]. For instance, events that precipitate negative emotional states, such as couple conflict, have been found to synchronize HRV and to elevate inflammatory compounds [19]. Physiological synchrony during such conflicts have also predicted marital dissatisfaction [20]. Similarly, synchrony of electrodermal activity (EDA), the difference in electrical potential between different areas of the skin, was enhanced during negative rather than positive interactions between romantic partners [21]. Relatedly, among dyadic peers, physiological synchrony has been observed to be heightened during social interactions that involve partners who disliked each other [23]. These divergent findings necessitate further systematic investigation into physiological synchrony.

Although prior research has demonstrated that physiological synchrony is observed across different relational categories and emotional states, none have examined how the mere copresence of a dyadic partner from different relational categories (i.e., stranger, friend, and romantic partner) influences physiological synchrony spanning various emotional states. Since the presence of another individual has been found to sufficiently trigger mechanisms that drive the exchange of social information [24] and inadvertently influence one’s physiological arousal [9] and action representation [25,26], it is crucial that we investigate how social and emotional factors may moderate the effect of mere copresence on physiological synchrony. Extending the work of Reference [9], the present study followed a similar experimental paradigm that differed in several ways. First, instead of testing participants in threes, the participants in this study were tested in pairs. Second, we expanded the dyads beyond that of strangers to include pairs of friends and romantic partner which allowed us to compare the mere copresence effect across different types of dyads. Third, while [9] only compared between fearful and embarrassing emotional states, we decided to investigate six distinct emotional states. This enabled us to discern the nuanced differences in levels of arousal that exist between emotional states. For example, fear is a negative emotional state that is accompanied by higher levels of arousal while sadness is associated with comparably lower levels of arousal. The present study sought to systematically investigate how the copresence of partners from different relational categories (i.e., strangers, companions (friends), and romantic partners (lovers)) and emotional states (i.e., embarrassment, sadness, fear, calmness, romance, and pride) influence physiological synchrony during a minimal social setup that did not require face-to-face communication. To this aim, we obtained an index of HRV, IBI, which reflects the regulatory activity of the ANS in response to emotions and is known to be directly related to vagal firing rate, such that a higher IBI index is indicative of greater parasympathetic response and, conversely, reflective of lower sympathetic arousal [27]. Consequently, physiological synchrony that measures similarity in the temporal coordination of IBI gleans an insight into the synchronisation of emotional arousal between dyadic partners in different emotional states. With respect to its two novel components of categories of relationships and emotional states, the present study has two sets of hypotheses. First, we expected that the effect of mere copresence on physiological synchrony would be positively correlated with relationship closeness so that lovers should exhibit the highest level of physiological synchrony, followed by friends and strangers (i.e., lovers > friends > strangers). Second, we expected to observe an effect of the type of relationship on physiological synchrony specific to emotional states. Similar to Reference [9], we hypothesised that strangers would exhibit synchrony in fearful and embarrassing situations. Given the contradictory findings in the present literature, we expected friends and lovers to synchronize their autonomic arousal during both positive and negative emotional states.

## 2. Materials and Methods

### 2.1. Participants

Participants were recruited among the students of the University of Trento (Italy) through public announcements on social networks and mailing lists. Inclusion criteria were age >18 years, heterosexual, Italian nationality, and with no history of medical or developmental conditions. A total of 124 participants took part in this study. The average ages for the 62 female and 62 male participants were 21.65 (SD = 2.77) and 23.48 years old (SD = 5.57), respectively. Participants were distributed in 62 opposite-sex pairs of friends (23 pairs), romantic partners (20 pairs), or strangers (19 pairs). No information about duration of the relationship and intimacy was collected from the pairs of friends and lovers. Participants were required to provide informed consent before the commencement of the study. Each participant was subsequently awarded university credits following the completion of the study. The study was conducted in accordance with the Declaration of Helsinki, at the University of Trento (Italy).

### 2.2. Procedure

In each experimental session, a male–female pair viewed a series of video clips while seated together side-by-side in front of the screen used to present the video stimuli. The subjects were required to avoid communication (verbal and nonverbal) during the video presentation. Data from the dyads that did not meet the experimental requirements were excluded from the experiment.

Romantic couples and friends signed up for the study together; each partner from the pair of strangers was recruited separately and was subsequently paired with a stranger of the opposite sex. All experimental sessions consisted of male–female pairs of participants.

Upon arriving at the laboratory, participants were instructed about the purpose of the study and signed the informed consent.

Each participant’s cardiac activity was recorded using an Electrocardiogram (ECG) sensor (FlexComp, Thought Technology). The ECG signal was measured throughout the entire presentation of 6 emotional videos. The experimental session lasted 30 min in total.

### 2.3. Stimuli

In a pilot study, 10 participants responded to a forced-choice single-answer questionnaire where they were instructed to pick one emotion from a list of six emotions that best represented each of 20 video clips. Beginning with 20 videos, we eventually selected six video clips that consistently elicited the same basic emotion across all participants. Each video clip was carefully screened for its ability to elicit one of six key emotions (i.e., embarrassment, sadness, fear, calmness, romance, and pride). Every participant was exposed to six 4-min video clips from different popular films or TV series that were used as the main stimuli for this study. To mitigate the possibility that a gory scene from the “The Walking Dead” clip might leave participants feeling uncomfortable if viewed last, we fixed the order in which the clips were presented. Specifically, the sequence of stimuli and order of presentation was as follows:A scene from the movie “When Harry met Sally” was used to elicit the emotion of embarrassment (EMBARRASS);A scene from the movie “Titanic” was used to elicit the emotion of sadness (SAD);A scene from the TV series “The Walking Dead” was used to elicit the emotion of fear (FEAR);A scene of a beach with a relaxing music playing in the background was used to induce calmness (CALMNESS);A scene from the movie “Notting Hill” was used to elicit romantic love (ROMANCE);A scene from the penalty-kick session in the 2006 FIFA World Cup Finals which led to the win of the Italian football team. Since all the participants were of Italian nationality, this stimulus was used to elicit the emotion of pride (PRIDE).

Before the start of each video clip, participants were presented with a 10-s image depicting the title of the video clip (on a white background) which they were about to watch. At the end of the last clip, a set of instructions would appear on the screen to inform participants that the session had ended. There was an interval of 1 min between the presentations of each video clip, where participants were exposed to an image of a white fixation point on a green background. The entire session lasted approximately 30 min.

### 2.4. Physiological Measures

Participant’s cardiac activity was assessed using a 3-electrodes ECG placed on the chest. Two ECG electrodes were placed between the left inferior area of the neck and mid-sagittal area of the left collarbone. The third electrode was placed near the lowest left rib area.

The R peaks corresponding to heart beats were detected from the ECG signal (Figure 1A, step S2) after it was first filtered (band pass filtering, cutoff frequencies: 10–48 Hz) to remove noise (Figure 1A, step S1). The result of the automatic detection is manually inspected for missing beats or misdetections and corrected to obtain the Inter Beat Intervals series (IBI). To deal with the different number of beats in the male and female IBI series, the IBI series were resampled at 2 Hz and filtered (low pass filter, cutoff frequency: 0.04 Hz) to remove high-frequency components of heart rate variability and then standardised (Figure 1A, step S3). For each IBI series ($IBI_{i}$), a surrogate IBI series ($\tilde{IBI_{i}}$) was generated using the Iterative Amplitude Adjusted Fourier Transform (IAAFT) [28] smoothed with a moving average filter (length 5 s) (Figure 1B, step S4).

### 2.5. Synchrony Measures

Cross-correlation was the chosen metric to quantify the presence of common patterns in the IBI series of the two members of the dyad [9,29,30]. Cross-correlation measures the extent to which the two physiological signals co-vary [30,31,32], while also allows a non-perfect alignment between the two time series through a lag parameter to account for anticipations or delays of the physiological response of one member with respect to the other. In our case, we computed the cross-correlation with multiple lags, between -10 s and +10 s with intervals of 1 s, and took the maximum value as a measure of the physiological synchrony between the two IBI series. We refer to this measure as the maximal cross-correlation within a time shift of ±10 s.

However, it is critical to assess whether the measured synchrony is actually due to the effect of copresence, to the stimulus, or to other random contributions. To discriminate between these factors, we adopted the framework proposed by Golland and colleagues [9,29] and computed three sets of physiological synchrony measures (see Figure 1B):*Copresence synchrony* between the real IBI series of the male and female of the dyad, who watched the videos together: This synchrony measure is influenced by both stimulus and copresence, i.e., the effect of watching the stimulus with the other member of the dyad. The copresence synchrony is computed between all the dyads of each group.*Stimulus synchrony* between the real IBI series of a male and a female belonging to different dyads who did not watch the videos together: This synchrony measure is therefore only influenced by the stimulus, as the male and the female did not watch the stimulus together. The stimulus synchrony is computed between all possible male–female pairs of each group, excluding the real dyad.*Surrogate synchrony* between surrogate signals of a male and a female: This synchrony measure is not influenced by the stimulus or by the copresence and is used to compose the distribution of the null hypothesis that there is no effect of synchrony due to stimulus or copresence. The surrogate synchrony is computed between all possible male–female pairs of each group.

The final result is a set of three distributions of synchrony measures (copresence, stimulus, and surrogate synchrony) for each type of relationship and stimulus. For the analysis of the physiological signals and the computation of the physiological synchrony, we used custom scripts based on pyphysio [33,34], physynch [35], and PySiology [36].

### 2.6. Analysis Plan

Before investigating physiological synchrony, we validated the collected physiological data. To this aim, we performed a two-way Analysis of the Variance (ANOVA) to investigate the effects of gender, type of relationship, and their interaction on the physiological response of subjects in terms of Heart Rate (HR; the inverse of the average IBI).

We then considered the analysis of the computed physiological synchrony. The objectives of this study are as follows:To replicate and extend the results of Golland and colleagues [9] by investigating the effects of copresence between dyads of strangers with different elicited emotions;To investigate the association between type relationship and physiological synchrony;To investigate the influence of emotional state on physiological synchrony across different relationship types.
While pursuing these three objectives, we were also concerned with statistically assessing whether the computed synchrony measures were actually influenced by copresence, by stimulation, or by random factors.

For each stimulus and type of relationship, we then compared the distributions of the surrogate and stimulus synchrony measures to test the null hypothesis that there are no effects due to the stimulus; then, we compared the distribution of the copresence and stimulus synchrony measures to test the null hypothesis that there are no effects due to copresence. The significance of the differences in the distributions was assessed with the Mann–Whitney test, fixing the significance threshold to α=0.05. The Mann–Whitney test was used instead of the Wilcoxon test to deal with the different number of samples in the groups. A *p*-values correction for multiple hypotheses was applied using the Benjamini–Hochberg procedure.

## 3. Results

The results of the ANOVA to investigate the effects of gender and relationship on the HR showed an effect of gender for all stimuli, with the exception of CALMNESS (F(1,116) = 3.77, p=0.055; females: M = 0.798, SD = 0.102; males: M = 0.840, SD = 0.130) and PRIDE (F(1,116) = 3.43, p=0.066; females: M = 0.776, SD = 0.104; males: M = 0.816, SD = 0.129). No significant effects of type of relationship or of the interaction between type of relationship and gender was found. The differences due to gender are expected and can be explained by physiological differences in emotional responses found between men and women in general [37].

We focused then on the investigation of the physiological synchrony between the dyads and in particular on the effects of the different type of relationships on the synchrony associated to the stimulus and to the copresence (see Figure 2 and Table 1).

We observed an overall effect of the stimulus on the physiological synchrony for all stimuli except for FEAR. In addition, strangers showed no stimulus effect for EMBARRASS and PRIDE and friends showed no stimulus for CALMNESS. It is interesting to note that strangers never show an effect of the stimulus on the three stimuli that are expected to elicit more arousal: EMBARRASS, FEAR, and PRIDE. On the other hand, for FEAR and PRIDE, we found a significant effect of copresence for strangers. We could interpret this result with an autonomic response more oriented to maximizing the interpersonal synchrony than to the actual responses to the stimulation.

We also found a significant effect of copresence for lovers to the stimulus CALMNESS and for friends to the stimulus ROMANCE.

However, the number of significant results might be penalized by the limited sample size and complexity of the experimental design (3 types of relationships, 6 stimuli, and 2 comparisons). To give a more complete landscape and to shed more light to the interpretation, it is useful to employ a less conservative approach and to consider an overall picture, with the other statistical results which were rejected by the multiple hypotheses correction.

Regarding strangers, we observed effects of stimulus and copresence for FEAR, CALMNESS, and ROMANCE. SAD showed an effect due to the stimulus but not to copresence, and PRIDE showed only an effect due to copresence. Similar results were also reported by the study we aimed to replicate, [9]. Notably, for the video for EMBARRASS, which was also used in Reference [9], neither stimulus nor copresence had an effect on synchronization. We addressed this discrepancy in regard to differences in the shorter duration of the stimuli (about 240 s in our pipeline) and to the different types of social group (Reference [9] tested strangers in groups of threes, whereas we tested pairs).

Overall, we can conclude that, for strangers, physiological synchrony occurs for different types of emotions, although some might cause a stronger response.

In general, while we found an effect of the stimulus (EMBARRASS, SAD, ROMANCE, and PRIDE for friends and lovers; FEAR for friends only; and CALMNESS for lovers only), the effects of copresence are found only for two videos (ROMANCE for friends and CALMNESS for lovers). This finding suggests that, when there is a relationship between the members of the dyad, synchrony is reduced in social situations which do not require direct interaction.

To further investigate this, we compared the distribution of the copresence synchrony between the three relationship group for all the stimuli (see Table 2).

Although none of the results survived the correction for multiple hypotheses, we noted that, in all the three videos where a *p*-value <0.05 is found (FEAR, PRIDE, and CALMNESS), Strangers have higher synchrony than lovers (FEAR and PRIDE) and friends (CALMNESS).

## 4. Discussion

We come into contact with numerous people in our daily lives, some of whom are strangers with whom we walk side-by-sid, but others are friends or romantic partners with whom we share most of our personal lives. Physiological synchrony underlies some of the social affiliative processes that forges and maintains these relationships [11,15,16,17,18]. It is therefore important to understand how the mere presence of others and our relationships with them affect us at the autonomic physiological level. Our study systematically investigated how physiological synchrony occurs in the copresence of dyads who are romantic couples, friends, or strangers when induced with different emotional states. Our first hypothesis that synchrony would be enhanced as a function of relationship closeness (i.e., lovers > friends > strangers) was not fulfilled. On the contrary, we found that strangers exhibited greater synchrony than lovers and friends across several emotional states. Second, we expected that the type of relationship and emotional state will influence the extent of physiological synchrony. Our hypothesis that the mere copresence of strangers would be associated with physiological synchrony in fearful and embarrassing emotional states was partially fulfilled. With some notable differences, we replicated the results in Golland and colleagues [9] and showed that copresence of strangers elicited synchrony in fearful situations. However, instead of exhibiting synchrony in embarrassing situations, synchrony was instead observed when pride was induced. We also hypothesised that lovers and friends would exhibit synchrony in an array of positive and negative emotional states. This hypothesis was partially fulfilled as the copresence of lovers and friends were related to positive emotional states of calmness and romance, respectively.

Less physiological synchrony in preexisting social relationships (i.e., between pairs of lovers and friends) compared to strangers might be due to the absence of novelty posed by the copresent individual in established relationships. Intuitively, partners in existing relationships ought to have shared similar emotions, which should be reflected in physiological synchrony [38]. However, these habitual physiological patterns might possibly render the autonomic arousal in lovers and friends more resistant to the influence of the mere presence of their partners. Having had prior experience in sharing physical space with a companion or romantic partner, these dyads may have less need to be alert to the social cues of their partner or to establish any immediate social connection with each other. While previous studies have reported that the mere presence of another person automatically influences mechanisms activated to drive the sharing of information [24,25,26], the copresent individual in such cases is a stranger to the participant and is therefore a novel social agent with whom the participant may instinctively be driven to consolidate a social bond [39]. Novelty represents a critical variable that induces alertness, excitement, and autonomic arousal [6], which might underlie greater synchrony observed between strangers at the autonomic level.

In the absence of a preexisting social relationship, synchronization of physiological arousal emerges between strangers, especially in fearful and prideful situations. The former emotion, fear, initiates the fight-or-flight response in the individual, which is paramount for survival [40]. Matching of autonomic arousal under conditions that are perceived to be threatening bears a close resemblance to a coordinated physiological response known as “physiological linkage.” Physiological linkage is widely displayed by social mammals and is presumed to present an evolutionary advantage (i.e., organised response) that enhances the odds of survival [11] and facilitates reproduction [41]. Consequently, matching of physiological arousal during threatening situations might be an evolutionarily conserved mechanism that facilitates a coordinated course of action that promotes the survival of the group and the individual. Conversely, physiological synchrony that emerges during the state of pride might reflect an attempt for strangers to coordinate their autonomic responses to initiate affiliation [39]. The stimulus used to elicit feelings of pride enforces the saliency of in-group belonging, that is, sharing a common nationality, which in turns facilitates prosocial behavior [42]. Numerous examples from ethnic rituals to military drills attest that synchronization of physiological arousal among strangers occur when a sense of group belonging is made prominent [43,44,45]. Supporting this interpretation, Jackson and colleagues [46] showed that, upon engaging in the same activity and placed in close proximity to each other, strangers exhibit a natural tendency to synchronize behaviors and levels of physiological arousal. All things considered, physiological synchrony may represent a potential mechanism by which social reciprocity between strangers is established [47].

Finally, physiological synchrony that is present in lovers in the calm emotional state might be due to co-regulation aimed to maintain similar homeostatic baselines. There is considerable evidence to show that romantic relationships serve as important social regulators of baseline homeostasis, including sleep patterns [48] and emotional arousal [49]. Thus, inducing a state of calm between romantic partners might have elicited synchronised co-regulation of physiological arousal that is typically present in the couple. In contrast, physiological synchrony in pairs of friends was observed during the romantic emotional state instead. To interpret this finding, we draw upon human evolutionary theories that underpin the psychology of opposite-sex friendships. Opposite-sex friends may serve as potential mates [50], and friendships with the opposite gender may provide greater potential for sexual access [51]. The closely intertwined psychology of opposite-sex friendships and human mating adaptations suggests that opposite-sex friendships may be seen as “back-up mates” and may indeed develop into mating opportunities. In line with this, we surmised that the synchrony observed in opposite-sex friends in our study may have been elicited in response to the stimulus that surfaces the notion of romantic affiliation. However, the specific reason underlying physiological synchrony remains to be discerned. For example, the matching of physiological states between platonic friends who do not harbour any attraction for each other could have reflected a state of alertness and discomfort in sharing a romantic moment, whereas friends who considered each other as potential mates might have experienced mutual excitement instead. Physiological arousal in the presence of an opposite-sex friend might have been made more pronounced in our sample which comprised college students who fall within the active “partner-seeking” phase of their lives [52]. As there is limited literature regarding this subject matter, future studies are required to disambiguate the physiological responses experienced by friends in such romantic situations.

This study has some limitations. First, we categorised dyads broadly into three main groups— companions, romantic partners, and strangers. In reality, not all couples within each of these groups function in the same way and subgroups of dyads may have different responses. For instance, the duration of relationship and extent of relationship satisfaction in romantic couples may have influenced the physiological synchrony observed [53]. Similarly, relationship closeness experienced in a friendship falls within a wide spectrum and may have had significant implications in modulating synchrony. Future studies should obtain behavioral measures regarding the characteristics of each relationship so as to better contextualise research findings.

Second, differences in personality constructs might have driven different physiological responses when viewing the series of video clips. Previous studies have found that the pairing of different personality traits within each dyad influences couple dynamics (e.g., Reference [54]) and could have also elicited unique patterns of synchrony that was not captured in the study.

Third, this study only investigated synchrony within a dyadic pair and further work is required to understand whether the same mechanisms are applicable to social groups, such as triads of lovers or friends. Fourth, our study utilized a minimal paradigm in which no direct face-to-face communication was required so as to investigate the effect of mere copresence of a stranger, romantic, partner, and companion. However, in the real world, verbal and nonverbal social signals inevitably occur between two people. For instance, friends tend to engage in more frequent nodding as compared to romantic partners, while the latter exhibit higher incidences of touch, mutual gazes, and silence [55]. Future investigation that allows for active interaction between members of a dyad would derive more ecologically valid findings.

Finally, there is also a limitation that should be noted in regard to the video stimuli used to elicit the emotions. The penalty-kick in the 2006 FIFA World Cup Finals that was used to induce a feeling of “pride” was the only clip that honed into the Italian nationality of our participants, whereas the other clips were from Hollywood movies. This could have potentially confounded feelings of “pride” with feelings of “belonging” due to an enhanced in-group salience. To account for this, the other video clips (i.e., embarrass and calmness) could also be specific to an Italian audience so as to diminish the group saliency effect. Alternatively, another video clip that does not enhance in-group salience based on nationality could be used to elicit feelings of “pride”.

## 5. Conclusions

As social beings, humans are dynamically influenced by our social interactions with others. The mere presence of another individual can affect us at a physiological level. The present study revealed several key findings regarding the influence of relationship type and emotional state on physiological synchrony of two copresent individuals. Compared to pairs of romantic partners and friends, the copresence of strangers led to greater physiological synchrony across several emotions. This finding sheds light to the notion of physiological synchrony that is currently wrought with discrepancies in the existing literature. At least during a passive social activity where no social interaction is required, our findings suggest that physiological synchrony is not necessarily reflective of relationship closeness between dyadic partners. Instead, physiological synchrony may emerge more strongly between strangers compared to dyads who are in existing relationships (i.e., friends and romantic partners). The absence of a preexisting relationship drives humans to synchronise their physiological responses to each other, potentially as a bid to establish social affiliation. Comparatively, in the presence of a friend or romantic partner with whom one has had prior social experiences with, there is less motivation to be alert to the social cues of the partner and to increase affiliation. Additionally, our study also revealed that strangers tend to synchronise their physiological arousal in situations of survival (i.e., when fear is induced) and when group belonging is made salient (i.e., when national pride is induced). Consequently, physiological synchrony may function as a key mechanism through which group survival and affiliative behaviours are precipitated. Applying this to the real world and inciting feelings of threat and group membership may produce synchrony at the physiological level that is likely to lead to a coordinated set of responses in individuals, even between strangers who are not acquainted with each other. Thus, from daily activities such as commuting to work to mass gatherings that advance social causes, we may exhibit greater similarity in our physiological patterns of responses with the strangers alongside us than previously thought.

## Figures and Tables

**Figure 1 behavsci-10-00011-f001:**
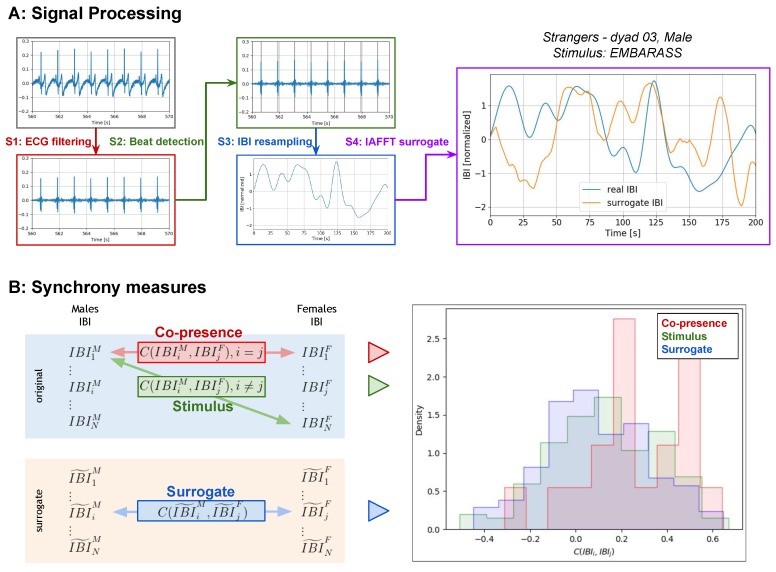
Data analysis: (**A**) Pipeline for the processing of the electrocardiogram (ECG) signal with the four main steps and the final result with the real and surrogate inter-beat interval (IBI) series signal for one subject and stimulus. (**B**) Three types of physiological synchrony and computation schemes, with the output distribution of the measures.

**Figure 2 behavsci-10-00011-f002:**
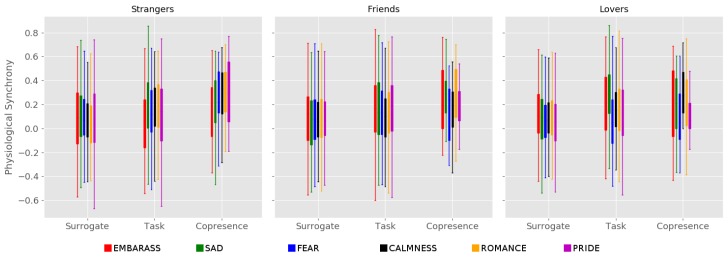
Distribution of the three types of physiological synchrony for each group of relationship and stimulus.

**Table 1 behavsci-10-00011-t001:** Results of the Mann–Whitney tests to compare between surrogate, stimulus, and copresence synchrony for each type of stimulus and type of relationship: Original *p*-values are reported. The results that remain significant after the correction for multiple hypotheses are in bold.

Emotion	Relationship	Surrogate vs Stimulus		Stimulus vs Copresence	
		U	*p*	U	*p*
EMBARRASS	Strangers	17341	*p* = 0.866	1313	*p* = 0.086
	Friends	30042	**p = 0.003**	2576	*p* = 0.182
	Lovers	15920	**p < 0.001**	1892	*p* = 0.488
SAD	Strangers	12707	**p < 0.001**	1571	*p* = 0.408
	Friends	27963	**p < 0.001**	2356	*p* = 0.066
	Lovers	10629	**p < 0.001**	2274	*p* = 0.926
FEAR	Strangers	14581	*p* = 0.046	1083	**p = 0.009**
	Friends	31537	*p* = 0.027	2663	*p* = 0.251
	Lovers	20119	*p* = 0.558	1869	*p* = 0.453
CALMNESS	Strangers	11565	**p < 0.001**	1215	*p* = 0.036
	Friends	32648	*p* = 0.098	2489	*p* = 0.126
	Lovers	16037	**p < 0.001**	1308	**p = 0.011**
ROMANCE	Strangers	11285	**p < 0.001**	1233	*p* = 0.043
	Friends	30897	**p = 0.011**	1972	**p = 0.005**
	Lovers	16660	**p = 0.002**	1584	*p* = 0.111
PRIDE	Strangers	14944	*p* = 0.094	1064	**p = 0.007**
	Friends	27676	**p < 0.001**	2760	*p* = 0.342
	Lovers	17289	**p = 0.011**	2054	*p* = 0.725

**Table 2 behavsci-10-00011-t002:** Results of the Mann–Whitney tests to compare the distribution of the copresence synchrony between the different relationship groups.

Emot.	Strangers vs Friends		Friends vs Lovers		Strangers vs Lovers	
	U	*p*	U	*p*	U	*p*
EMBARRASS	187	*p* = 0.217	218	*p* = 0.390	170	*p* = 0.292
SAD	190	*p* = 0.240	195	*p* = 0.200	183	*p* = 0.428
FEAR	176	*p* = 0.144	190	*p* = 0.168	120	*p* = 0.025
CALMNESS	149	*p* = 0.041	160	*p* = 0.045	188	*p* = 0.483
ROMANCE	214	*p* = 0.460	204	*p* = 0.267	174	*p* = 0.332
PRIDE	169	*p* = 0.108	170	*p* = 0.074	116	*p* = 0.019

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
