# Peer review of "Strangers, Friends, and Lovers Show Different Physiological Synchrony in Different Emotional States"

_behavsci, 2019, doi:10.3390/bs10010011_

Round 1

Reviewer 1 Report

Review

Strangers, friends, and lovers show specific physiological synchronies in different emotional contexts

Bizeggo et al.

This paper replicates and extends the work from Golland et al. (2015). It is important to replicate previous findings and I applaud the authors for their work. However, I have struggled while reading the paper and I think the paper needs serious reorganization and revision.

The title is a bit puzzling to me, let me explain why. The phrase “specific physiological synchronies” to me suggests that you have measured a variety of different physiological responses, however, you focus only on HRV. To me “different emotional contexts” suggest different setting that elicit emotions, i.e. a classroom, a family setting or in a bar. However, the authors use the term ‘context’ to refer to different emotions. I don’t think this is the appropriate concept to refer to different videos that portray different emotions.

The organization of the manuscript didn’t work for me. I’ve read the manuscript in the order that was provided, so I’ve read the results section without having read the methods section. It is very hard to understand the results without having read about the methods. I don’t understand why the author’s have chosen this order. I’ve looked up some similar papers in the journal and they have a regular order of introduction, methods, results, discussion, conclusion. I suggest that the manuscript is rewritten using the regular order.

In general, the introduction was rather short and lacked motivation for why the study was conducted. I think the focus of replicating and extending Golland et al. should be foregrounded. Related to that it wasn’t clear to me how the current study differed from previous work. In addition, the set-up of the co-presence in the study didn’t become clear.

You present a lot of tables, I wonder whether they all contribute to your story, in particular the long Tables with mainly null-results.

With regard to the discussion I missed the discussion of the different emotions and the heterogeneous findings for the different emotion. Also I was wondering what would happen in face-to-face interaction/communication settings? What is the implication of their findings?

Below I will list more detailed comments.

Line 40: incompatible results between HRV and SC could be due to the fact that SC is only linked to the parasympathetic system, while HRV is linked to both systems.
Line 46: “emotional context” I don’t think this is a good term to refer to different emotional situations or emotional states.
Line 48: please explain to your reader why you are interested in the autonomic arousal. And does your audience know what this means?
Line 51: “the extent of convergence of emotional responses” what does this mean? How was this measured? This isn’t clear to the reader yet.
Line 53: “co-presence” how does this work in your set-up without face-to-face communication?
Line 56-58: Your second hypothesis seems to contradict the results from [15], how did you end up with this hypothesis?

Line 62: EE: what is this construct? How did you compute this?
Line 64: 3-way ANOVA? Context*relationship*gender
Line 70: this sentences is ungrammatical
Line 73: I found this result a bit misleading. You report significant results in table 2, but they don’t survive Bonferroni correction. So, they shouldn’t be marked in the table, in fact I think it’s better not to mention them at all. After all, it’s a non-significant result and should be treated likewise.
Line 74: you report a series of 2-way ANOVA’s, why didn’t you do pairwise comparisons? And what does a higher EE mean?

Figure 1: the legend is unclear, U=S

Table 1: the results should all be given using the same number of decimals, it’s a bit sloppy this way.

Figure 2: very small and what is on the y-axis? I would also consider to place the figure later in the manuscript, only after 2.3 where is should be discussed (which it isn’t really now).

Line 82: what is IBI?
Line 86, 87: the results are given in table 3 and 4 already and are not significant so redundant.
Line 92: be careful not to confuse absence of evidence with evidence of absence!

Line 94, 95: “The stimuli are appropriate to elicit different emotions” what is this conclusion based on? And similar “.. gender have no effect” but there were two significant effects in EE and 4 for IBIB for gender, how does this fit with this conclusion?

Table 5: caption: typo co-presence. Why are the N for Surrogate and Stimulus a factor 10 larger than co-presence?

Line 102-106: what do these results mean?
Line 103, 104: copresence or co-presence? Inconsistent
Line 109,110: this claim is too strong, it’s not completely independent of type of elicited emotion, there were differences between the emotions.
Line 118: “engage” suggests an active role of the participants, but as far as I’m aware they were passively viewing the videos.
Line 121: Figure ?

Figure 3: is redundant, the results do not survive correction so shouldn’t be introduced like this in a figure.

Line 130: “affect us at the most basic physiological level” why is this relevant?
Line 185: main conclusion is missing

Line 198: “pairviewed”? And how were the participants seated, what was their task?
Line 232: what were the results of the manipulation check?

Figure 4: I think you should reverse A and B since you first discuss the B panel.
Line 246: Step S3, is that where you compute the surrogate? And in general, what is the function of the surrogate? This is also missing in the results and discussion.

Line 261: which analyses did you do? Be more precise please.
Line 270: “effects of being with the member of the dyad” what does this mean?
Line 279: why did you o a Mann-Whitney test?

Author Response

This paper replicates and extends the work from Golland et al. (2015). It is important to replicate previous findings and I applaud the authors for their work. However, I have struggled while reading the paper and I think the paper needs serious reorganization and revision.

The title is a bit puzzling to me, let me explain why. The phrase “specific physiological synchronies” to me suggests that you have measured a variety of different physiological responses, however, you focus only on HRV. To me “different emotional contexts” suggest different setting that elicit emotions, i.e. a classroom, a family setting or in a bar. However, the authors use the term ‘context’ to refer to different emotions. I don’t think this is the appropriate concept to refer to different videos that portray different emotions.
Thanks for the suggestion. We changed the title to:
“STRANGERS, FRIENDS, AND LOVERS SHOW DIFFERENT PHYSIOLOGICAL SYNCHRONY IN DIFFERENT EMOTIONAL STATES”

The organization of the manuscript didn’t work for me. I’ve read the manuscript in the order that was provided, so I’ve read the results section without having read the methods section. It is very hard to understand the results without having read about the methods. I don’t understand why the author’s have chosen this order. I’ve looked up some similar papers in the journal and they have a regular order of introduction, methods, results, discussion, conclusion. I suggest that the manuscript is rewritten using the regular order.
The structure of the manuscript has been revised. Now the organization follows the regular order: Introduction, Materials and Methods, Results, Discussion and Conclusions

In general, the introduction was rather short and lacked motivation for why the study was conducted. I think the focus of replicating and extending Golland et al. should be foregrounded. Related to that it wasn’t clear to me how the current study differed from previous work. In addition, the set-up of the co-presence in the study didn’t become clear.

Thank you for this comment. We have made extensive changes to the Introduction section and have foregrounded the study by Golland et al. We have also specified how the present study differed from that of Golland et al. Moreover, we have clarified our focus on co-presence in the introduction.

You present a lot of tables, I wonder whether they all contribute to your story, in particular the long Tables with mainly null-results.
We removed many parts that did not contribute substantially to the content of the manuscript.

With regard to the discussion I missed the discussion of the different emotions and the heterogeneous findings for the different emotion. Also I was wondering what would happen in face-to-face interaction/communication settings? What is the implication of their findings?We have revised the Discussion section extensively, adding interpretations and implications regarding the findings on specific emotions. We have also included the need to investigate face-to-face communication in the limitations section. We have also summarised the implications of our findings in the conclusion section.

Below I will list more detailed comments.

Line 40: incompatible results between HRV and SC could be due to the fact that SC is only linked to the parasympathetic system, while HRV is linked to both systems.
Thanks for the comment. We included this explanation in the manuscript.

Line 46: “emotional context” I don’t think this is a good term to refer to different emotional situations or emotional states.
Thanks for the suggestion. We preferred the use of “emotional state”, or “stimulus” when referring to the experimental design

Line 48: please explain to your reader why you are interested in the autonomic arousal. And does your audience know what this means?
We rephrased the sentence to highlight that we are interested in investigating the regulatory activity of the Autonomic Nervous System in response to emotions.

Line 51: “the extent of convergence of emotional responses” what does this mean? How was this measured? This isn’t clear to the reader yet.

We have rephrased this line and explained the association between physiological synchrony and similarity of emotional experiences.

Line 53: “co-presence” how does this work in your set-up without face-to-face communication?

We have clarified the minimal set-up paradigm in the Methods section and explained how participants were told to refrain from verbal and non-verbal communication. Co-presence thus only involves two people being in the presence of each other without direct communication.

Line 56-58: Your second hypothesis seems to contradict the results from [15], how did you end up with this hypothesis?
Thanks for the comment. Our hypothesis was actually more general, as not stated in the manuscript.

Line 62: EE: what is this construct? How did you compute this?
This is related to the Emotional Embedding part that has been removed.

Line 64: 3-way ANOVA? Context*relationship*gender
This is related to the Emotional Embedding part that has been removed.

Line 70: this sentences is ungrammatical
This is related to the Emotional Embedding part that has been removed.

Line 73: I found this result a bit misleading. You report significant results in table 2, but they don’t survive Bonferroni correction. So, they shouldn’t be marked in the table, in fact I think it’s better not to mention them at all. After all, it’s a non-significant result and should be treated likewise.
This is related to the Emotional Embedding part that has been removed.

Line 74: you report a series of 2-way ANOVA’s, why didn’t you do pairwise comparisons? And what does a higher EE mean?
This is related to the Emotional Embedding part that has been removed

Figure 1: the legend is unclear, U=S
This is related to the Emotional Embedding part that has been removed.

Table 1: the results should all be given using the same number of decimals, it’s a bit sloppy this way.
This is related to the Emotional Embedding part that has been removed.

Figure 2: very small and what is on the y-axis? I would also consider to place the figure later in the manuscript, only after 2.3 where is should be discussed (which it isn’t really now).
Thanks for the comment. We created a different figure and placed it into the Results section.

Line 82: what is IBI?
We restructured the manuscript and not IBI is defined in the Materials and Methods section

Line 86, 87: the results are given in table 3 and 4 already and are not significant so redundant.
The table has been removed.

Line 92: be careful not to confuse absence of evidence with evidence of absence!
We thank the reviewer for pointing this out and apologize for the methodological error. We removed the sentence

Line 94, 95: “The stimuli are appropriate to elicit different emotions” what is this conclusion based on? And similar “.. gender have no effect” but there were two significant effects in EE and 4 for IBIB for gender, how does this fit with this conclusion?
The sentence was indeed not clear. In addition, we removed the parts relative to the Emotional Embedding. The sentence has been removed.

Table 5: caption: typo co-presence. Why are the N for Surrogate and Stimulus a factor 10 larger than co-presence?
Thanks for the correction, we fixed the typo. Following the suggestions to reduce the number of figures and tables we moved the table in the supplementary material as the same information is graphically presented in Figure 2.

The N for copresence (Ncopresence) is the number of male-female pairs for each group.
For Stimulus and Surrogates, instead is the number of all the possible male - female pairings for each group. Thus:
Nstimulus = Ncopresence * (Ncopresence - 1) / 2 (all pairings excluded the real dyad)
Nsurrogate = Nstimulus + Ncopresence (all pairings included the real dyad)

Line 102-106: what do these results mean?

We have included a more thorough explanation of our main findings in the Discussion section

Line 103, 104: copresence or co-presence? Inconsistent
Thanks, we fixed the typo.

Line 109,110: this claim is too strong, it’s not completely independent of type of elicited emotion, there were differences between the emotions.
We agree that the sentence was not fully supported by the results. We rephrased it.

Line 118: “engage” suggests an active role of the participants, but as far as I’m aware they were passively viewing the videos.
The sentence was misleading. “Engaged” referred to the existence of a relationship between the two members of the dyad. The sentence has been corrected

Line 121: Figure ?
We removed the typo

Figure 3: is redundant, the results do not survive correction so shouldn’t be introduced like this in a figure.
We removed the figure

Line 130: “affect us at the most basic physiological level” why is this relevant?

We removed this phrase and added provided further explanation as to the relevance of measuring the effect of co-presence on physiological synchrony

Line 185: main conclusion is missing

We added the main conclusions in the Discussion and the Conclusion sections

Line 198: “pairviewed”? And how were the participants seated, what was their task?
Thanks for the comment. We fixed the typo and provided more details about the experimental procedure.

Line 232: what were the results of the manipulation check?
The manipulation check was part of the data collected for another study. For this study it was only considered to compute the Emotional Embedding. Consequently to the removal of all the Emotion Embedding related contents, we also removed this sentence from the manuscript.

Figure 4: I think you should reverse A and B since you first discuss the B panel.

We implemented the suggestion of the reviewer and edited the figure to make it more clear.

Line 246: Step S3, is that where you compute the surrogate? And in general, what is the function of the surrogate? This is also missing in the results and discussion.
We rewrote the description of the signal processing procedure, which was not clear. In particular, the reference to step S3 is now correctly assigned to the creation of the surrogate series. The function to generate the surrogate series is the Iterative Amplitude Adjusted Fourier Transform algorithm, as described in the referenced paper (Schreiber et al. 2000).
We also split the subsection into Physiological Measures and Synchrony Measures, to better clarify.

Line 261: which analyses did you do? Be more precise please.
The Analysis Plan subsection has been thoroughly revised to better specify the statistical analyses.

Line 270: “effects of being with the member of the dyad” what does this mean?
We modified the sentence to clarify.

Line 279: why did you o a Mann-Whitney test?
The Mann-Whitney test was used instead of the Wilcoxon test to deal with the different number of samples in the groups. We added this sentence to clarify.

Reviewer 2 Report

The present manuscript explores dyadic physiological synchrony across emotional contexts and relationship types. First, an evolutionary perspective on social bonds in terms of relative advantages of forming interpersonal relationships is taken. Moreover, the concept of embodiment is introduced and an adequate definition of synchrony is given. In providing the current status of literature including contradictory findings, the need for additional research is conclusively described. The authors aim to investigate the influence of relational categories and emotional context on physiological synchrony, namely heart rate variability, controlling for direct interaction. Results show that values of emotional embedding (EE) differed among the types of stimuli (emotional context in videos). Stimulus synchrony – physiological coupling in dyads exposed to the same video but not watching together - was found across emotional contexts in strangers (FEAR, CALMNESS, ROMANCE, SAD), friends (FEAR, ROMANCE, SAD, EMBARRASS, PRIDE) and lovers (CALMNESS, ROMANCE, EMBARRASS, SAD). Further, co-presence synchrony (i.e. synchrony between participants watching videos together) was found in strangers (FEAR, CALMNESS, ROMANCE, PRIDE) rather than friends (ROMANCE only) and lovers (CALMNESS only). Subsequently, central elements of the study with respect to the methodology (e.g. sample characteristics, measures, procedure and statistical analysis) are outlined in the materials and methods section, after the discussion section. Finally, the authors discuss results and conclude that the absence of a pre-existing relationship, such as in strangers, is associated with more physiological synchrony.

The manuscript covers a promising topic. However, there are several issues that need to be addressed:

Introduction

Core topics mentioned in the introduction provide well-founded evidence regarding the role of physiological synchrony in research on relationship type and emotion. However, instead of only touching important findings briefly, it would be better to explain them in greater detail. In addition, we suggest to cover further studies such as Coutinho et al., 2019, Helm et al., 2011, Kleinbub, 2017, Tschacher & Meier, 2019 as well as early studies of Levenson and Gottman (1983; 1985). The authors mention the autonomic nervous system (ANS) regulating heart rate resp. heart rate variability in a few words. However, the paper would benefit from a more profound explanation of ANS activation and what it stands for. Additional information to its ambiguous physiological meaning could shed light on previous contradictory findings. The Karvonen study (15) is cited as evidence for higher EDA synchrony of romantic partners in negative interactions, but this is not what was reported by Karvonen et al. This is rather the finding of Coutinho et al. (2019) published recently in the journal Family Process.

Materials and Methods

The sequence of sections is unconventional. To support the flow of reading, the methods section should follow right after the introduction. Authors computed the cross-correlations of an "IBI series". Yet, during the 4-minute video stimuli, the male participant commonly has fewer IBIs than the female -- thus, the IBI series becomes desynchronized. How did authors deal with this? "average IBI" as a variable is the same as "heart rate". Authors may point to this. If the focus is on IBI series, it is unclear how the authors studied heart rate variability (HRV), or if they did this at all. HRV is claimed in the abstract.In the participant section, group characteristics in terms of age and gender are nicely described. However, it is not clear how individuals were recruited and whether any inclusion/exclusion criteria were applied. The computational strategy of synchrony represents the core element of the presented findings. We would encourage the authors to give more explanation about the rationale of computation of synchrony as maximal cross-correlation within a window of ± 10 seconds. There might have been just a single random peak in the cross-correlations, an outlier, and no other systematically significant cross-correlations at all -- it is dubious whether this counts as evidence of synchrony. Statistical procedures used to address the proposed research questions are mentioned in the results. However, description and reasoning behind statistical models (e.g. ANOVA) should already be outlined in the analysis plan. In addition, the analysis plan including stage one and two would benefit from revision. We suggest to reorganize its structure based on relationship type and emotional structure rather than findings of previous studies. Furthermore, these should be used to explain results in the discussion section. The EE measure is rather a measure of "affective intensity", if I understand your PCA right. You should rename this measure, also to avoid misunderstandings (the 6 videos do actually provide the "emotional embedding", right?) Too little information is provided concerning the experimental setup: were participants allowed to communicate (nonverbally or verbally) during the video presentations? Did they communicate (even if not allowed)? How were they seated? This information is essential for making sense of the condition "presence".

Results

Although illustration of results is well recommended, the authors should reduce the number of plots and tables focusing on the most important findings. Moreover, some layouts should be revised (e.g. horizontal bars indicating significance in figure 3). Labels in figure 1 are not clear and not explained in the legend. We suggest reporting of findings according to the structure in the methods part.

Discussion

The authors explain why physiological synchrony was more consistently present across emotional contexts in strangers rather than romantic couples and friends. However, interpretation and implications of results with respect to different emotional contexts are missing. In fact, we think the future directions section should be expanded.

References

Few errors in terms of font style as well as details on page numbers should be corrected.

In conclusion, the current manuscript is of interest to interpersonal synchrony research. However, in order to enhance the quality and understanding, the manuscript needs to be revised. We suggest accepting the paper following major revision.

Author Response

The manuscript covers a promising topic. However, there are several issues that need to be addressed:

Core topics mentioned in the introduction provide well-founded evidence regarding the role of physiological synchrony in research on relationship type and emotion. However, instead of only touching important findings briefly, it would be better to explain them in greater detail. In addition, we suggest to cover further studies such as Coutinho et al., 2019, Helm et al., 2011, Kleinbub, 2017, Tschacher & Meier, 2019 as well as early studies of Levenson and Gottman (1983; 1985). The authors mention the autonomic nervous system (ANS) regulating heart rate resp. heart rate variability in a few words. However, the paper would benefit from a more profound explanation of ANS activation and what it stands for. Additional information to its ambiguous physiological meaning could shed light on previous contradictory findings. The Karvonen study (15) is cited as evidence for higher EDA synchrony of romantic partners in negative interactions, but this is not what was reported by Karvonen et al. This is rather the finding of Coutinho et al. (2019) published recently in the journal Family Process.

Thank you for this comment. As for the core topics in physiological synchrony, we have decided to centre the Introduction on Golland et al, the study from which we drew our experimental paradigm. However, the other studies that were mentioned are still included in the Introduction. In the Introduction section, we have explained the link between autonomic arousal, homeostasis and emotional states. The ANS was also included in the Discussion section. We have also corrected the Karvonen study and replaced it with Countinho et al. (2019).

The sequence of sections is unconventional. To support the flow of reading, the methods section should follow right after the introduction.
The structure of the manuscript has been revised. Now the organization follows the regular order: Introduction, Materials and Methods, Results, Discussion and Conclusions

Authors computed the cross-correlations of an "IBI series". Yet, during the 4-minute video stimuli, the male participant commonly has fewer IBIs than the female -- thus, the IBI series becomes desynchronized. How did authors deal with this?
The resampling of the IBI series to 2 Hz (as described in Golland et al. 2015) allowed us to deal with this issue. We now explicitly highlighted that:
“The resampling step is needed to obtain an equal number of samples for the male's and female's series.”

"average IBI" as a variable is the same as "heart rate". Authors may point to this.
After the restructuring of the manuscript, “average IBI” appears only once, to explain its relationship with the Heart Rate.

If the focus is on IBI series, it is unclear how the authors studied heart rate variability (HRV), or if they did this at all. HRV is claimed in the abstract.
To quantify the physiological synchrony, we compute the cross-correlation between the male’s and female’s IBI series. As a result, we expect higher synchrony when the IBI series have similar patterns of variations. In this sense, we think that it is correct to say that we study the Heart Rate Variability (HRV), or, better, how the HRV of the two members of the dyad evolve in synchrony.
We considered not to change the manuscript.

In the participant section, group characteristics in terms of age and gender are nicely described. However, it is not clear how individuals were recruited and whether any inclusion/exclusion criteria were applied.
Thanks for the comment. We added this information in the Participants subsection.

The computational strategy of synchrony represents the core element of the presented findings. We would encourage the authors to give more explanation about the rationale of computation of synchrony as maximal cross-correlation within a window of ± 10 seconds.
Thanks for this comment. The maximal cross-correlation is the core metrics within the computational framework proposed by Golland and colleagues (Golland et al. 2014). We expanded the text to explain how the cross-correlation was used and added more references of other studies which used maximal cross-correlation to measure synchrony of physiological signals (Mass et al 2005, Kettunen et al 2000a, Kettunen et al. 2000b).

There might have been just a single random peak in the cross-correlations, an outlier, and no other systematically significant cross-correlations at all -- it is dubious whether this counts as evidence of synchrony.
We are aware that the cross-correlation metrics is subject to errors due to signal artifacts and spurious components.
However, while an erroneous measure due to a random peak might indeed occur, we disagree that this can be systematic across all subjects so as to invalidate the measure.
Rather, it is exactly to address this issue that we adopted a robust analysis plan where we compare the distribution of the synchrony measures from real signals and from surrogate signals.
If our results would come from random peaks or spurious components, these would be present in both real and surrogate signals and no significant difference could be found between the two distributions.
Finally, to mitigate the risk of wrong results, we manually inspected the IBI series to correct beat detection errors and outliers that could compromise the signal.

Statistical procedures used to address the proposed research questions are mentioned in the results. However, description and reasoning behind statistical models (e.g. ANOVA) should already be outlined in the analysis plan.
The Analysis Plan subsection has been thoroughly revised. The statistical analyses are now mentioned there.

In addition, the analysis plan including stage one and two would benefit from revision. We suggest to reorganize its structure based on relationship type and emotional structure rather than findings of previous studies. Furthermore, these should be used to explain results in the discussion section.
We reorganized the Analysis Plan and  restructured the manusctript. We dismissed the organization of the Analysis Plan into Stage 1 and Stage 2.
We feel that now the manuscript is more clear and easy to follow.

The EE measure is rather a measure of "affective intensity", if I understand your PCA right. You should rename this measure, also to avoid misunderstandings (the 6 videos do actually provide the "emotional embedding", right?)
This is related to the Emotional Embedding part that has been removed.

Too little information is provided concerning the experimental setup: were participants allowed to communicate (nonverbally or verbally) during the video presentations? Did they communicate (even if not allowed)? How were they seated? This information is essential for making sense of the condition "presence".
Thanks for highlighting the lack of this essential information. We added more information about the experimental procedure.

Although illustration of results is well recommended, the authors should reduce the number of plots and tables focusing on the most important findings.
We removed many parts that did not contribute substantially to the content of the manuscript.

Moreover, some layouts should be revised (e.g. horizontal bars indicating significance in figure 3).
Following the suggestion of reviewer 1 [R1.30], we decided to remove Figure 3

Labels in figure 1 are not clear and not explained in the legend.
The Figure has been removed from the manuscript as related to the discussion on the Emotional Embedding.

We suggest reporting of findings according to the structure in the methods part.
We thoroughly revised the structure of the manuscript and reported the result according to the Analysis Plan in the Materials and Methods section

The authors explain why physiological synchrony was more consistently present across emotional contexts in strangers rather than romantic couples and friends. However, interpretation and implications of results with respect to different emotional contexts are missing. In fact, we think the future directions section should be expanded.

We have included a more thorough interpretation of our findings in regard to the different emotional contexts. Future directions have also been suggested to verify our postulations.

Reviewer 3 Report

The introduction section should be a little more theory-driven. Although the introduction of the paper defines synchrony and its association with HRV, and differences among relational categories, little rationale is given to “emotional context” defined by the authors. In fact, I suggest the authors to reconsider their choice of words. Those reported by the authors seem to be emotional states people tend to experience according to the context and interaction with social factors. I suggest authors also to explore more the conceptualization and association among relational categories and emotional states.

Line 230-233 is where I have major concerns. Authors state using a 7-point Likert scale on

“wether each video was unpleasant/pleasant, scary/funny, embarrassing/non-embarrasing”. However, those emotional states reported by the authors do not correspond fully to the emotional states under analysis. For example, how can authors use current items to assess “calmness” using items measuring  “scary/funny”? As a matter of fact, scary and funny are not dichotomous factors, thus, they should not be used as opposite range points at any given item. Additionally, are those 3 items created by the authors or have they been validated in previous literature? The authors are encouraged to give some more thoughtful consideration about what they measured and how they measured.

I think the paper could benefit if, in the discussion, the meaning of the results is explained more thoroughly. Furthermore, there is an insufficient explanation of the theoretical and practical consequences of the findings, or of the specific ways in which they contribute to the literature on the topic.

Specific comments:

Line 188: why is it important to reference heterosexual participants? Could homosexual pairs influence current results? The authors are recommended to present a thoughtful rationale.

Line: 216-224: The authors report different videoclips from renowned movies and series to elicit distinct emotional states. However, the authors used a specific moment in the history of football, the penalty-kick in the 2006 FIFA World Cup Finals between Italy and France to elicit the emotion pride, in which Italy won the World Cup. This seems to bias results since the readers do not know and authors do not report participants' citizenship nor nationality. Please explain and provide rationale.

Line 254: acronyms should be revised. For example, Emotional Embedding (EE) is first referenced in Line 62, however, its explanation only appears afterward.

Author Response

The introduction section should be a little more theory-driven. Although the introduction of the paper defines synchrony and its association with HRV, and differences among relational categories, little rationale is given to “emotional context” defined by the authors. 

We have extensively revised the Introduction and provided a more theory-driven introduction in regard to the autonomic nervous system and the emotional contexts.

In fact, I suggest the authors to reconsider their choice of words. Those reported by the authors seem to be emotional states people tend to experience according to the context and interaction with social factors.
Thanks for this comment, which was also raised by another reviewers. Folloing the suggestion we used the terms “emotional states” or “stimuli” when referring to the experimental design

I suggest authors also to explore more the conceptualization and association among relational categories and emotional states.

We thank you for this comment and have re-structured our Introduction to present a better conceptualisation of relational categories and emotional states.

Line 230-233 is where I have major concerns. Authors state using a 7-point Likert scale on
“whether each video was unpleasant/pleasant, scary/funny, embarrassing/non-embarrasing”.
However, those emotional states reported by the authors do not correspond fully to the emotional states under analysis. For example, how can authors use current items to assess “calmness” using items measuring  “scary/funny”? As a matter of fact, scary and funny are not dichotomous factors, thus, they should not be used as opposite range points at any given item. Additionally, are those 3 items created by the authors or have they been validated in previous literature? The authors are encouraged to give some more thoughtful consideration about what they measured and how they measured.
The 7-point likert scale data were collected for another study. For this study it was only considered to compute the Emotional Embedding. Consequently to the removal of all the Emotion Embedding related contents, we also removed this sentence from the manuscript with any other mention to the likert scale.

I think the paper could benefit if, in the discussion, the meaning of the results is explained more thoroughly. 

The meaning of the results have been explained more thoroughly in the Discussion section.

Furthermore, there is an insufficient explanation of the theoretical and practical consequences of the findings, or of the specific ways in which they contribute to the literature on the topic.

We have illuminated the implications and contributions of our findings in the Discussion and Conclusion sections. 

Line 188: why is it important to reference heterosexual participants? Could homosexual pairs influence current results? The authors are recommended to present a thoughtful rationale.

In the discussion, we have included evolutionary theories that point to the psychological mechanisms of opposite-sex friends which have been developed in the context of heterosexual mating. Therefore, we thought that it is important to reference heterosexual participants.

Line: 216-224: The authors report different videoclips from renowned movies and series to elicit distinct emotional states. However, the authors used a specific moment in the history of football, the penalty-kick in the 2006 FIFA World Cup Finals between Italy and France to elicit the emotion pride, in which Italy won the World Cup. This seems to bias results since the readers do not know and authors do not report participants' citizenship nor nationality. Please explain and provide rationale.
Thanks for highlighting this issue. We added more details to the description of the stimulus to explain why it was used to elicit pride.

Line 254: acronyms should be revised. For example, Emotional Embedding (EE) is first referenced in Line 62, however, its explanation only appears afterward.

We have revised this so that the explanation appears after the acronym

Round 2

Reviewer 2 Report

The issues raised in my initial review were satisfactorily covered.

Author Response

Thanks for the feedback

Reviewer 3 Report

I would like to commend the authors on the revision of the manuscript and thank them for the point-by-point response to all reviewers. The paper is greatly improved. There are a couple of minor points that could do with further clarification and increase study quality.   

The authors explained the use of the penalty-kick in the 2006 FIFA World Cup Finals to elicit pride, however, while the clarification seems assertive, it is also associated with a strong limitation. Aren’t there any other videos that could have elicit EMBARRASS, CALMNESS, etc. specifically in Italian participants? I hope the authors understand this question, as it does could bias results. This limitation should be acknowledged in the limitations section, in addition to a suggestion for how this limitation could be addressed in future research (e.g., using other “neutral” videos to elicit pride).

Please revise references. Some are incomplete, mostly the new inserted ones.

Specific comments:

Line 296: authors report “[?]”, thus, authors are recommended to provide reference.

Author Response

We thank the reviewer for the comment and we edited as suggested the limitation section as follow

... Future investigation that allows for active interaction between members of a dyad would derive more ecologically-valid findings. Finally, there is also a limitation that should be noted in regard to the video stimuli used to elicit the emotions. The penalty-kick in the 2006 FIFA World Cup Finals that was used to induce a feeling of “pride” was the only clip that honed into the Italian nationality of our participants, whereas the other clips were from Hollywood movies. This could have potentially confounded feelings of “pride” with feelings of “belonging” due to an enhanced in-group salience. To account for this, the other video clips (i.e. embarrass, calmness) could also be specific to an Italian audience so as to diminish the group saliency effect. Alternatively, another video clip that does not enhance in-group salience based on nationality could be used to elicit feelings of “pride”.

we also fixed the references

We hope the MS is now ready

The authors